# Electrochemistry Studies of Hydrothermally Grown ZnO on 3D-Printed Graphene

**DOI:** 10.3390/nano9071056

**Published:** 2019-07-23

**Authors:** Dimitra Vernardou, George Kenanakis

**Affiliations:** 1Center of Materials Technology and Photonics, School of Engineering, Hellenic Mediterranean University, 710 04 Heraklion, Crete, Greece; 2Institute of Electronic Structure and Laser, Foundation for Research and Technology-Hellas, N. Plastira 100, Vasilika Vouton, GR-700 13 Heraklion, Crete, Greece

**Keywords:** 3D-printed graphene, zinc oxide, hydrothermal growth, electrochemical studies

## Abstract

A three-dimensional (3D) printer was utilised for the three-dimensional production of graphene-based pyramids and an efficient hydrothermal procedure for ZnO growth. In particular, the 3D-printed graphene pyramids were forwarded in Pyrex glass bottles with autoclavable screw caps filled with 50 mL of an aqueous solution of zinc nitrate hexahydrate and hexamethylenetetramine for 1 h at 95 °C; sufficient enough time to deposit well-dispersed nanoparticles. X-ray diffraction patterns were in accordance with a Raman analysis and presented the characteristic peaks of graphite along with those of wurtzite ZnO. Different positions on the sample were tested, confirming the uniform dispersion of ZnO on graphene pyramids. From the electrochemical studies, it was found that the charging and discharging processes are affected by the presence of ZnO, indicating one well-defined plateau for each process compared to the previously reported bare graphene pyramids. In total, the material shows a value of 325 mAh g^−1^, a capacitance retention factor of 92% after 5000 scans, and a coulombic efficiency of 100% for the first scan that drops to 85% for the 5000th scan. This excellent performance is the result of the effect of ZnO and graphene that combines two Li^+^ accommodation sites, and the contribution of graphene pyramids, which provides more available sites to favor lithium storage capacity. Hence, this anode may be a promising electrode material for lithium-ion batteries.

## 1. Introduction

Metal oxides have been widely investigated as electrodes for Li-ion batteries due to their high theoretical capacities, low cost and natural abundance [1,2].In particular, ZnO has received increased attention as a large band gap semiconductor with distinctive photocatalytic, sensing and electrochemical properties [3,4,5]. As an anode material, it has a particularly high theoretical capacity of 978 mAh g^−1^ (i.e., higher than that of graphite, which is 370 mA h g^−1^) [6], chemical stability and a lack of safety issues associated with its use. Nevertheless, achieving a high rate capability and good stability has been very challenging because of its low electronic conductivity and undesirable large volume change during its charging and discharging processes [7,8,9,10]. To tackle this drawback, one proposed strategy is to decrease the particle size of the metal oxide (i.e., to the nanoscale level). This will provide a larger area of active material, accommodating effectively the strain of Li intercalation/deintercalation in nanomaterials [11,12,13]. However, low conductivity and inferior cyclability remain critical issues limiting its application [14,15].

The exceptional conductivity, abundant adsorption sites and short diffusion paths associated with graphene are expected to further enhance the transport processes within the oxide nanomaterials [16]. A major advantage of graphene over other carbon materials is the excellent distribution of metal oxides on its surface [17]. The advantages of ZnO/graphene composites were evaluated in many studies [18] estimating a reversible capacity of 515 mAh g^−1^ after 100 cycles at 200 mA g^−1^ [19], a high initial discharge capacity of 1583 mAh g^−1^ at 200 mA g^−1^ [11] and a reversible capacity of 653.7 mAh g^−1^ after 100 cycles [20].

In this work, ZnO is deposited on three-dimensional (3D)-printed graphene [21] using the environmentally friendly hydrothermal process [22] at 95 °C for 1 h. The hydrothermal procedure and the printing technique are both considered to be low-cost and easily compatible with the production of electrodes with new shapes and compositions. Considering these aspects, the proposed material may make an important contribution to the fabrication of battery material technology.

## 2. Materials and Methods

Commercially available PLA (polylactic acid)-graphene filaments (purchased from BlackMagic3D, Ronkonkoma, NY, USA) with a volume resistivity of 0.6 ohm⋅cm, compatible with typical FDM (Fused Deposition Modeling) 3D printers, were used. As reported elsewhere [21], the development of graphene pyramids was carried out using a 0.4 mm nozzle at 240 °C, while the surface where the object was printed was adjusted at 50 °C. The 3D printing parameters (nozzle speed, extrusion velocity, layer thickness, printing path, etc.) were optimized in order to obtain uniformly printed samples with a filling ratio of 100%.

The 3D-printed PLA-graphene pyramids (as presented in [21]) were forwarded in Pyrex glass bottles with autoclavable screw caps filled with 50mL of a 0.01 M, aqueous solution of Zn(NO_3_)_2_·6H_2_O and HMTA at 95 °C. The HMTA acts asa reducing agent in the hydrothermal synthesis of ZnO [23]. In particular, it decomposes upon heating to form formaldehyde and ammonia, which then react with H_2_O to produce OH^−^ and promote the precipitation reaction. Since the oxidation state of Zn in ZnO and Zn(NO_3_)_2_·6H_2_O is 2, the role of the reducing agent is in the nucleation and growth rate processes of ZnO materials instead of the reduction ofZn^2+^ ions [24]. After 1 h of deposition time, one could notice a white ZnO layer on the 3D-printed pyramids [22,23,25]. The composites were then washed with MilliQ H_2_O, dried in air at 95 °C, and characterized using the techniques described below. The growth temperature was carefully controlled with thermometers inside the oven and the Pyrex glass bottles during the whole growth period.

The surface morphology of the samples was determined using a field emission scanning electron microscope (FE-SEM, JEOL JSM-7000F, Tokyo, Japan) with a magnification of 100×.The crystal structure of all ZnO/3D-printed graphene samples was studied by X-ray diffraction (XRD, Rigaku (RINT 2000 diffractometer, Tokyo, Japan) with Cu Kα (λ = 1.5406 Å). Raman measurements were carried out using a Horiba LabRAM HR (Kyoto, Japan) via a 532 nm solid state laser with an output power of 100 mW [21].

Electrochemical analysis of the materials was performed in an electrochemical cell [26] using Pt, Ag/AgCl and the ZnO/3D-printed graphene as the counter, reference and working electrode, respectively, in 1M LiCl for a scan rate of 10 mV s^−1^ at potential ranging from −1.0 V to +0.5 V. Galvanostatic measurements were obtained in the range −0.5 V to +0.5 V under a specific current of 40 mA g^−1^.

## 3. Results

### 3.1. Surface Morphology

Figure 1 depicts a typical SEM photograph of a ZnO/3D-printed graphene pyramid. The3D structure of the printed samples can be clearly seen in Figure 1 to approach a truncated pyramid instead of an ideal square pyramid. Moreover, one can see from Figure 1 the layer-by-layer fabrication of the 3D-printed samples, with an ~100 μm spatial resolution in the z-axis, which is a typical resolution for most FDM 3D printers.

### 3.2. Structure

A typical XRD pattern of ZnO/3D-printed graphene samples is presented in Figure 2, verifying their crystalline properties. It depicts the characteristic peaks of the PLA profile [27], along with the graphite (JCPDS card, No.75-1621 [28,29]), indicating the existence of multilayered graphene in a PLA matrix. Finally, the XRD pattern of ZnO is identically matched with the characteristic peaks of ZnO’s hexagonal P6(3)mc structure, in agreement with JCPDS card file No. 36-1451 and the literature [30].

A typical Raman spectrum of ZnO/3D-printed graphene samples is presented in Figure 3. The key features of the Raman spectrum at around 1584 and 1356 cm^−1^ are the characteristic ones for carbon materials, the so-called G and D peaks. According to the literature associated with Raman analysis of graphene [31,32], the peak at 1584 cm^−1^ (E_2g_ mode (G band) of graphite) is related to the vibration of sp^2^-bonded carbon atoms. Additionally, the peak at 1356 cm^−1^ is associated with the vibration of carbon atoms (D band). Accordingly, the phonon frequencies of the Raman spectra attributed to ZnO are: 332 cm^−1^ (multiple-phonon scattering processes), 379 cm^−1^ (A1(TO)), 438 cm^−1^ (E2(high)), 483 cm^−1^ (2LA), 574 cm^−1^ (A1(LO)), and 663 cm^−1^ (TA+A1(LO)); which are in agreement with the literature [30,33,34,35], with the E2(high) mode at 438 cm^−1^ exhibitingthe strongest intensity. The rest peaks shown in the Raman spectrum of Figure 3 can be matched to the PLA of the materials. The Raman peak at 1046 cm^−1^ can be assigned to the C–CH_3_ stretching mode, whereas the one at 871 cm^−1^ to the C–COO stretching of the polymer unit [36,37]. The samples were studied in different positions indicating similar spectra(as indicated in the inset of Figure 3), revealing the characteristic Raman peaks for graphene (G and D bands), along with the Raman fingerprints of PLA and ZnO, confirming the uniform dispersion of ZnO on 3D-printed graphene pyramids.

### 3.3. Electrochemical Studies

Figure 4 indicates the cyclic voltammograms for 5000 scans sweeping the potential from −1.0 V to +0.5 V versus Ag/AgCl at 10 mV s^−1^. The voltammogram for the first scan shows two cathodic humps at −0.60 V/−0.19 V and two respective anodic humps at −0.54 V/+0.05 V. Additionally, it may be seen that the anodic humps shift to higher potentials, whereas the cathodic humps to lower values after 2500 scans. Nevertheless, the curves are stable and overlap in subsequent scans, indicating that the material presents high electrochemical reversibility and stability after a prolonged period. The high resistivity of the cyclic voltammograms may be due to the low conductivity of the electrolyte solution or the graphene pyramids. Work on the electrochemical study of the material is currently in progress using a higher concentration of electrolyte solution and developing graphene pyramids on a conductive substrate.

The chronopotentiometric (CP) curves of the ZnO/3D-printed graphene for the first and the 5000th scan at 40 mA g^−1^ and a potential range of −0.50 V to +0.50 V are shown in Figure 5a. It may be observed that the charging and discharging processes are affected by the ZnO’s presence, indicating one well-defined plateau for each process as compared with the bare graphene [21]. Additionally, the material shows a specific discharge capacity of 325 mAh g^−1^ and a capacitance retention factor of 92% after 5000 scans (Figure 5b). This excellent performance is due to the effect of ZnO and graphene that combines two Li^+^ accommodation sites [18]. In particular, the well-dispersed ZnO on the 3D-printed graphene network offers an additional redox site (apart from graphene; C element). Furthermore, the contribution of graphene pyramids provides high surface areas and more available sites to facilitate the Li-storage capacity. The reduction of ZnO into Zn and the formation of a lithium–zinc alloy are proposed regarding the Li-redox reactions in the active sites of ZnO [38,39]. Accordingly, the electroactive sites in graphene pyramids for Li storage can be expressed according to the equation xLi^+^ + C_6_ + xe^−^ <-> Li_x_C_6_ [21]. In Table 1 below, the specific discharge capacity value and scan number of other anode materials are shown and compared with the one reported in this work. It is indicated that the anode material that was prepared by the combination of a 3D-printing technique and a hydrothermal procedure has not been reported before utilising either an organic or an inorganic electrolyte. The estimated capacity value after 5000 scans is promising for further exploitation in Li-ion batteries if one considers the low cost and the high safety of the materials technology along with the aqueous-based electrolytes, respectively.

The anode material exhibited a coulombic efficiency of 100% that dropped to 85% after 5000 scans as shown in Figure 5b. The coulombic efficiency observed in this material is much higher than that of ZnO@GN anodes (82.1%) [18], SiO_2_@C hollow spheres (≈45%) [43] and carbon/ZnO nanorod array anodes (44%) [44], showing an excellent reversible capacity. Rate capability was also evaluated at specific currents of 40 mA g^−1^, 80 mA g^−1^, 100 mA g^−1^, and 200 mA g^−1^ as presented in Figure 5c. The anode maintains a high value of 290 mAh g^−1^ under a specific current of 200 mA g^−1^. Furthermore, it delivered a value of 322 mAh g^−1^ when the specific current returned to 40 mA g^−1^, suggesting high reversibility of the electrode.

## 4. Conclusions

Hydrothermally grown ZnO on 3D-printed graphene was accomplished at 95 °C for 1 h. The materials were crystalline and the ZnO was uniformly dispersed on graphene pyramids. The presence of both ZnO and 3D-printed graphene pyramids affected the electrochemical behavior of the anode, indicating a specific discharge capacity of 325 mAh g^−1^ and a capacitance retention factor of 92% after 5000 scans. This excellent performance is due to the combination of two Li^+^ accommodation sites from ZnO and graphene along with the contribution of graphene pyramids facilitating more available sites. Considering these outcomes, the proposed anode material may significantly contribute to the fabrication of battery materials technology.

## Figures and Tables

**Figure 1 nanomaterials-09-01056-f001:**
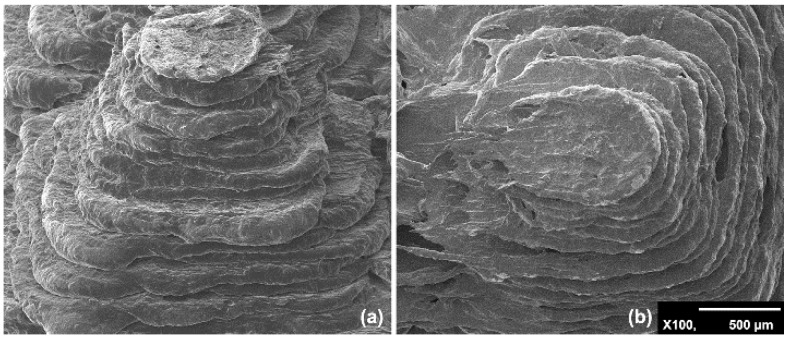
Typical SEM photographs of the ZnO/three-dimensional (3D)-printed graphene pyramids: (**a**) side view; (**b**) top view.

**Figure 2 nanomaterials-09-01056-f002:**
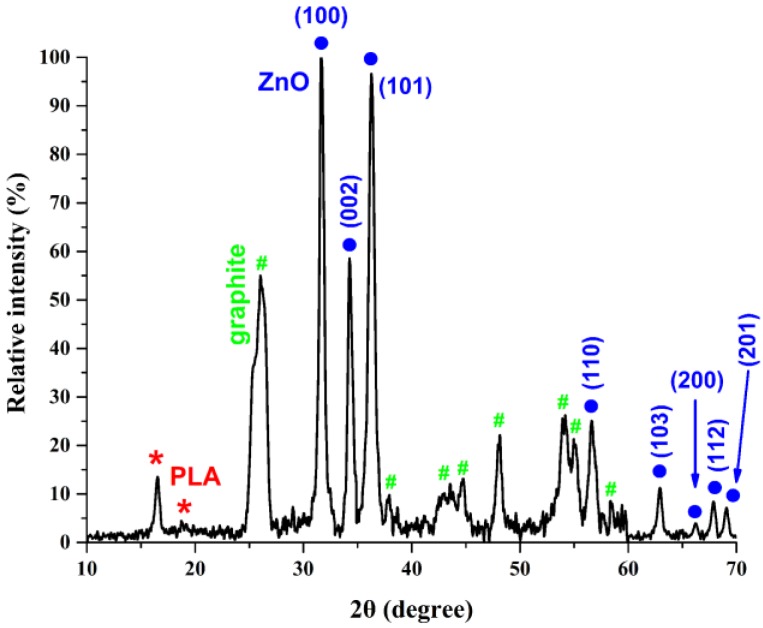
A typical XRD pattern of ZnO/3D-printed graphene samples. The XRD peaks of the graphite, PLA and ZnO are indicated with green hashes, red asterisks and blue dots, respectively.

**Figure 3 nanomaterials-09-01056-f003:**
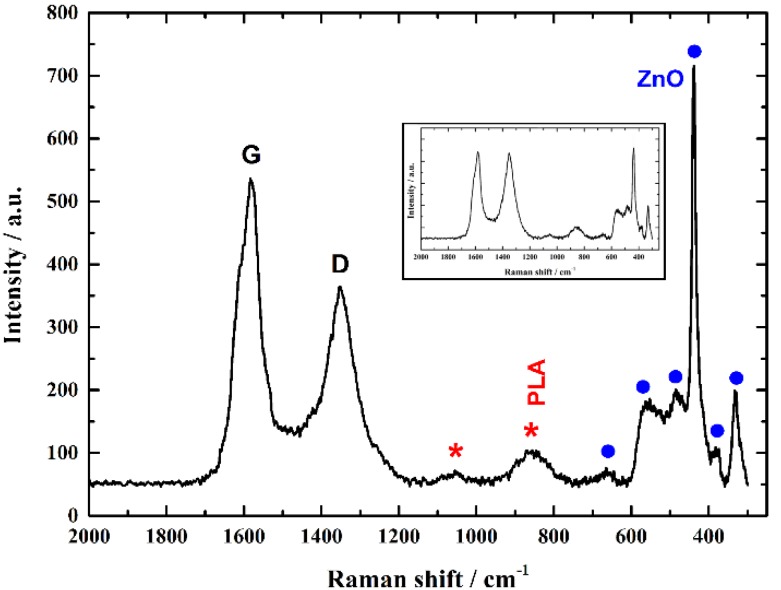
A typical Raman spectrum of ZnO/3D-printed graphene samples (PLA matrix and ZnO labeled with red asterisks and blue dots, respectively). In the inset of Figure 3, one can see a Raman spectrum at a different position of the same 3D-printed sample, indicating the uniform dispersion of ZnO on 3D-printed graphene pyramids.

**Figure 4 nanomaterials-09-01056-f004:**
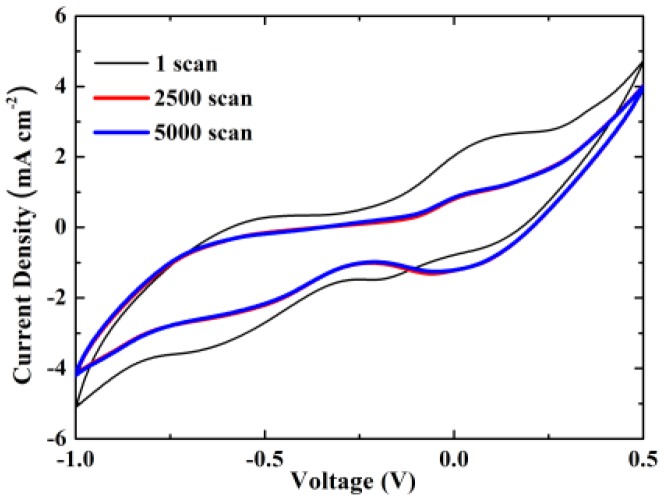
Cyclic voltammograms of ZnO/3D-printed graphene for 10 mV s^−1^ and a number of scans up to 5000.

**Figure 5 nanomaterials-09-01056-f005:**
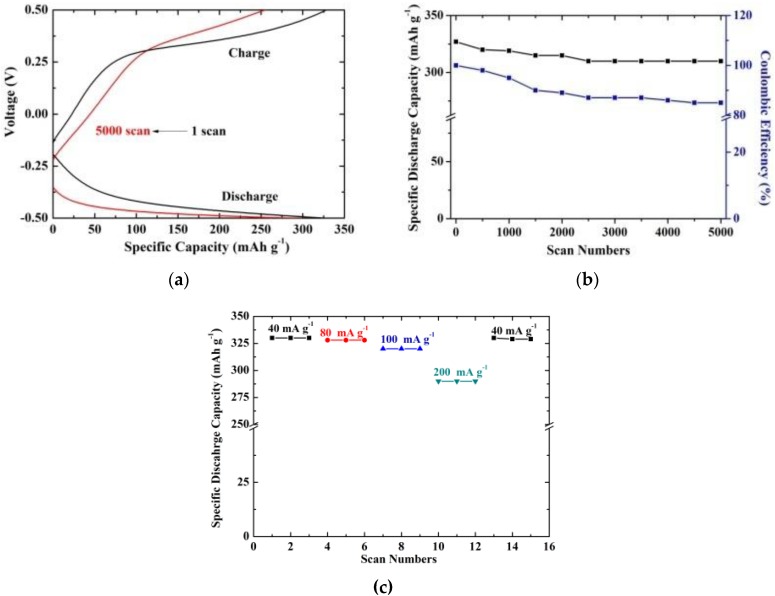
(**a**) The chronopotentiometric (CP) curves of ZnO/3D-printed graphene for −0.5 V to +0.5 V at 40 mA g^−1^ (the charge and discharge curves are indicated along with the 1st and the 5000th scan in black and red color, respectively); (**b**) Specific discharge capacity and coulombic efficiency of the anode material as a function of the scan numbers; (**c**) Rate capability of the composite at 40, 80, 100, 200 mA g^−1^ and then back to 40 mA g^−1^.

**Table 1 nanomaterials-09-01056-t001:** Coulombic efficiency and specific discharge capacity of anodes taken from the literature.

Anodes	Specific Discharge Capacity	Scan Number	References
ZnO-CoO-C	438 mAh g^−1^	50 (0.0–2.5 V)	[40]
ZnO/graphene	250 mAh g^−1^	100 (0.005–3.0 V)	[41]
ZnO-Loaded/Porous carbon composite	512.7 mAh g^−1^	10 (0.1–3.0 V)	[20]
ZnO nanorod	358 mAh g^−1^	30 (0.3–3.0 V)	[42]
3D-printed graphene	265 mAh g^−1^	1000(−1.0–0.5 V)	[21]
ZnO/3D-printed graphene pyramids	306 mAh g^−1^	5000 (−1.0–0.5 V)	This work

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
