# Peer review of "Electrochemistry Studies of Hydrothermally Grown ZnO on 3D-Printed Graphene"

_nanomaterials, 2019, doi:10.3390/nano9071056_

Reviewer 1 Report

The paper describes the synthesis of a ZnO-graphene material to be used as electrode in a Li ion battery. After resubmission, the paper quality has been improved, even though some points should be clarified prior to be published

- The authors claim that HMTA acts as reducing agent in the hydrothermal synthesis of ZnO. Starting from a salt where Zn presents also +2 as oxidation state, could you explain which specie is being reduced?

- Regarding the XRD characterization of the ZnO/3D-printed graphene:

* the XRD patterns of pure ZnO and graphene material should be included or referred with comparative purposes

* If we are talking about a graphene material, why the authors discuss the “characteristics peaks of graphite”?

- About the Raman analysis, what happens with the second order spectra? Did the authors measure the typical band of graphene in this frequency range?

- In my opinion, the main lack of the manuscript is the CV characterization. What do you think about the high resistivity of the CVs shown? What happens with the humps when increasing the scan rate?

- In figure 5a) each charge/discharge profile should be identified (including, for example, a legend)

Author Response

Dear Reviewer 1,

Thank you very much for your response regarding our manuscript entitled:

“Electrochemistry studies of hydrothermally grown ZnO on 3D-printed graphene”

we wish to publish in Nanomaterials under the Special Issue of Nano Carbon for Batteries Applications. We have proceeded with the second revision of our manuscript and we are resubmitting our work hoping that we have fully complied with your recommendations. The highlighted revision encloses all changes in bold.

Reviewer’s comments:

The paper describes the synthesis of a ZnO-graphene material to be used as electrode in a Li ion battery. After resubmission, the paper quality has been improved, even though some points should be clarified prior to be published.

Thank you very much for your comments. Please see our response to your comments below.

- The authors claim that HMTA acts as reducing agent in the hydrothermal synthesis of ZnO. Starting from a salt where Zn presents also +2 as oxidation state, could you explain which specie is being reduced?

Response: Thank you for the comment. It is general accepted from the literature that HMTA provides continuous source of OH- to promote the precipitation process. Since, the oxidation state of Zn in ZnO and Zn(NO3)2·6H2O is 2, the role of the reducing agent is the nucleation and growth rate processes of ZnO materials instead of reducing Zn2+ to Zn or Zn1+ ions.

We have proceeded with the respective changes in our revised manuscript including also a reference to support our argument.

*the XRD patterns of pure ZnO and graphene material should be included or referred with comparative purposes

Response:  Thank you for the comment. As already reported, the XRD peaks of the graphite, PLA and ZnO, are indicated with green hashes, red asterisks, and blue dots, respectively.

The XRD patterns of pure ZnO are identically matched with the characteristic peaks of ZnO hexagonal P6(3)mc structure (in agreement with JCPDS card file No. 36-1451, and Ref [29] of our manuscript).

Moreover the diffraction peaks of pure graphene are perfectly matched with the peaks of graphite (JCPDS card, No.75-1621 [27,28]).

For brevity, we are not presenting the diffraction patterns of pure ZnO and graphene. Moreover, since we are talking about a commercial filament (please check our experimental details), we feel that such an information is not necessary.

In order to make it more clearly, we have rephrased our revised manuscript.

* If we are talking about a graphene material, why the authors discuss the “characteristics peaks of graphite”?

Response: Thank you for the comment. There is indeed a big confusion in the literature whether you have graphite or graphene. (This is actually the reason why Raman spectroscopy is used in order to verify the existence of graphene instead of graphite; The D and G peak shapes for graphite and graphene are different).

In our case, there is homogenously spread in to a polymer matrix. We are then talking about multilayered graphene, which actually exhibits the same XRD pattern with graphite. This is why we are discussing about XRD of graphite.

In order to make it more clear, we have rephrased our revised manuscript.

- About the Raman analysis, what happens with the second order spectra? Did the authors measure the typical band of graphene in this frequency range?

Response: Thank you for the comment. The samples were studied in different positions indicating similar spectra (as indicated in the inset of Figure 3), revealing the characteristic Raman peaks for graphene (G and D bands), along with the Raman fingerprints of PLA and ZnO, confirming the uniform dispersion of ZnO on 3D-printed graphene pyramids.

We have revised our manuscript accordingly.

- In my opinion, the main lack of the manuscript is the CV characterization. What do you think about the high resistivity of the CVs shown? What happens with the humps when increasing the scan rate?

Response: Thank you for the comment. We believe that the high resistivity of CVs may be due to the low conductivity of electrolyte solution and the graphene pyramids. These could be improved increasing the concentration of the electrolyte solution and developing graphene pyramids on a highly conductive substrate such as ITO. These considerations are currently under progress.

We have made the necessary changes in the revised manuscript to clarify this point.

As we mentioned in the first revision, the curves with different scan rates have been removed because they are not representative. Unfortunately, the measurements cannot be repeated because it has been passed a long time since they were performed.

- In figure 5a) each charge/discharge profile should be identified (including, for example, a legend)

Response: Thank you for the comment. In the revised manuscript, “charge” and “discharge” words were included along with the necessary changes in Figure 5 (a) caption to clarify the set of curves.

With our best regards,

Dimitra Vernardou, PhD and co-author

Reviewer 2 Report

I am satisfied with the changes made by the authors, and the manuscript can be accepted in the present version.

Author Response

Dear Reviewer 2,

Thank you very much for your response regarding our manuscript entitled:

 “Electrochemistry studies of hydrothermally grown ZnO on 3D-printed graphene”

 we wish to publish in Nanomaterials under the Special Issue of Nano Carbon for Batteries Applications.

With our best regards,

Dimitra Vernardou, PhD and co-author

Round  2

Reviewer 1 Report

Thank you for your response. 

This manuscript is a resubmission of an earlier submission. The following is a list of the peer review reports and author responses from that submission.

Round  1

Reviewer 1 Report

 In this work, the authors reported the synthesis of “ZnO/ 3D-printed graphene pyramids” nanocomposites and investigated the applications towards lithium storage properties. It is interesting to combine hydrothermal synthesis with 3D printing substrates. However, there are also some critical points and inconsistence need to be clarified. I will recommend it for publication after a major revision.

1. Page 2, in XRD pattern, besides the peaks belonging to ZnO and PLA, there are other diffraction peaks, which should be indexed.

2. Page 3, line 93-94, “The samples were studied in different position indicating similar spectra and confirming the uniform dispersion of ZnO on 3D-printed graphene pyramids”. The authors should give data to support.

3. Additional characterizations, especially SEM and TEM results, should be provided to check the morphology, size, and microstructures of the samples.

4. The weight percentage of ZnO or graphene in the composite should be determined, and the contribution of each component in the final lithium storage should be discussed.

5. Lithium storage part, the author should use coin cell configuration, including the organic electrolyte, lithium as counter/reference electrode, and active material/binder/conductive agent as working electrode, to study the lithium storage properties.

6. The authors should make a table to compare the lithium storage properties of the synthesized ZnO/3D graphene composites and other reported ZnO anodes.

7. Spelling and grammatical mistakes, repetition and other errors in the manuscript should be carefully checked.

Reviewer 2 Report

The paper describes the synthesis of a ZnO-graphene material to be used as electrode in a Li ion battery. Even though the topic is of interest between the scientific community, the manuscript presents some important drawbacks to be considered by the authors:

- The quality of English should be improved

- The abstract should be rewritten, remarking the preparation procedure and the final application

- Regarding the hydrothermal treatment, the authors should clarify the low temperature and time selected (surprisingly lower than commonly treatments previously published). Moreover, how do you ensure the proper experimental conditions by using glass bottles?

- What about the time/cost of 3D printing? Is this procedure useful thinking about scalability in the production of battery electrodes?

- Which is the role of HMTA in the formation of ZnO particles?

- The characterization of the material obtained is quite poor. SEM images should be included. Moreover, the discussion of XRD and Raman results should be improved

- The electrochemical characterization is also poor. For example, when considering the CVs, the authors talk about anodic/cathodic peaks while only oxidation/reduction humps are observed. Moreover, the CVs shown present a high resistivity which should be removed. When increasing the scan rate, the above mentioned humps disappear which evidence the bad rate capability.

Unfortunately, and mainly by the above described, I consider that this manuscript should be REJECTED